# Optimizing and Implementing a Community-Based Group Fall Prevention Program: A Mixed Methods Study

**DOI:** 10.3390/ijerph21020162

**Published:** 2024-01-31

**Authors:** Maaike van Gameren, Paul B. Voorn, Judith E. Bosmans, Bart Visser, Sanne W. T. Frazer, Mirjam Pijnappels, Daniël Bossen

**Affiliations:** 1Department of Human Movement Sciences, Faculty of Behavioural and Movement Sciences, Amsterdam Movement Sciences Research Institute, Vrije Universiteit Amsterdam, 1081 BT Amsterdam, The Netherlands; m.van.gameren@vu.nl (M.v.G.); p.b.voorn@hva.nl (P.B.V.); m.pijnappels@vu.nl (M.P.); 2Centre of Expertise Urban Vitality, Faculty of Health, Amsterdam University of Applied Sciences, 1105 BD Amsterdam, The Netherlands; b.visser2@hva.nl; 3Department of Health Sciences, Faculty of Science, Amsterdam Public Health Research Institute, Vrije Universiteit Amsterdam, 1081 BT Amsterdam, The Netherlands; j.e.bosmans@vu.nl; 4Consumer Safety Institute (VeiligheidNL), 1062 XD Amsterdam, The Netherlands

**Keywords:** accidental falls, health program, focus group, experiences, preventive intervention

## Abstract

Falls and fall-related injuries among older adults are associated with decreased health. Therefore, fall prevention programs (FPPs) are increasingly important. However, the translation of such complex programs into clinical practice lacks insight into factors that influence implementation. Therefore, the aim of this study was to identify how to optimize and further implement a widely used group-based FPP in the Netherlands among participants, therapists and stakeholders using a mixed methods study. FPP participants and therapists filled out a questionnaire about their experiences with the FPP. Moreover, three focus groups were conducted with FPP participants, one with therapists and one with other stakeholders. Data were analysed according to the thematic analysis approach of Braun and Clarke. Overall, 93% of the 104 FPP participants were satisfied with the FPP and 86% (n = 12) of the therapists would recommend the FPP to older adults with balance or mobility difficulties. Moreover, six themes were identified regarding further implementation: (1) recruiting and motivating older adults to participate; (2) structure and content of the program; (3) awareness, confidence and physical effects; (4) training with peers; (5) funding and costs; and (6) long-term continuation. This study resulted in practical recommendations for optimizing and further implementing FPPs in practice.

## 1. Introduction

Falls and fall-related injuries can have significant consequences for older adults [1].

About 30% of all adults 65 years or older experience at least one fall per year [2]. As a result, 10% are treated at the emergency department for fall-related injuries [3]. Besides physical injuries, falls can result in a fear of falling and reduced quality of life [4,5,6]. Moreover, falls are associated with high healthcare costs [3]. Due to the ageing population, the number of falls and related injuries is expected to increase even further [3,7].

Fall prevention programs (FPPs) have been shown to reduce the number of falls and injuries in adults of 65 years or older [8]. Many of these programs are based on exercise and differ in content and target group. Although the optimal characteristics of successful FPPs remain unclear, programs that incorporate multiple components (e.g., strength and balance) and contain high doses and challenging balance exercises seem to be more effective [8,9]. An example of such an FPP is the Otago Exercise Program (OEP) for community-dwelling older adults who can individually exercise safely [10]. This is a globally implemented intervention that contains individually tailored strength-, balance- and mobility exercises [10]. These exercises last 30 min and participants must exercise three times a week and are encouraged to walk outside at least twice a week [10].

Another example is the home-based eHealth intervention StandingTall [11]. This intervention consists of balance training via individually tailored exercise prescription using a tablet at home [11]. Participants are encouraged to exercise at least 120 min per week for 2 years [11]. A third FFP example is the Stepping On program, which aims to improve self-efficacy and includes exercises focusing on improving balance and strength and encouraging behavior that stimulates safety and mobility (such as emphasizing the influence of medication on the risk of falls and practicing mobility techniques in an outdoor location) [12]. Stepping On includes weekly two-hour sessions conducted for seven weeks [12]. Although the FFPs described above differ in content, they all have demonstrated efficacy.

Another FPP that is frequently used in the Netherlands is the ‘In Balance’ program [13], which is based on the core elements of fall prevention, such as multicomponent interventions including balance, functional and resistance exercises [14]. In Balance is a 14-week group FPP that aims to reduce falls by increasing awareness, balance and strength in adults 65 years or older with an increased risk of falling. The first four weeks comprise counselling and educational meetings once per week. These educational meetings cover topics regarding fall prevention to increase knowledge about preventing falls and increase awareness of fall risks and balance disruption. The last ten weeks comprise two one-hour exercise sessions per week. Exercises are derived from the principles of Tai Chi and are mainly focused on physical balance and strength. In Balance is provided by certified therapists, with a background in exercise and physiotherapy.

In 2006, In Balance was shown to be effective in reducing the number of fallers in pre-frail older adults living in care centres [13] and has since been implemented on a wide scale in the Netherlands. However, during the years, the target group shifted to independently living older adults. In addition, the program’s duration has been shortened from twenty to fourteen weeks because the largest physical improvements were seen in this initial period [13]. To evaluate this widely implemented, yet adjusted program, we are currently conducting a randomized, controlled trial with the aim to determine the (cost-)effectiveness of the In Balance program [15]. However, the implementation of such complex interventions is often difficult and results in an adaptation of the program. Therefore, insight into the implementation process provides context to research findings and may identify barriers and facilitators for the further implementation of the FPP and similar programs.

Identifying the experiences and perceptions of those involved is crucial in order to gain insight into how FPPs can be optimized and provides insights to aid dissemination and implementation [16]. Additionally, the results of this study can provide new insights for the implementation of other FPPs in a real-life setting [17]. Therefore, the aim of this study was to conduct a process evaluation among FPP participants, therapists and other stakeholders to identify how the In Balance FPP can be optimized and further implemented.

## 2. Methods

### 2.1. Study Design, Recruitment and Participants

The process evaluation is part of a project in which we investigate the (cost-)effectiveness of the In Balance program [15]. The recruitment of FPP participants and therapists was carried out via mail, email and telephone. For the stakeholder group, we invited people working at the municipal health services, municipality, elderly associations, Royal Dutch Society for Physical Therapy, Association of Cesar and Mensendieck Exercise Therapists, Consumer Safety Institute and health care insurance companies.

For the current study, we used a convergent parallel mixed methods design. This design involves the following: The quantitative and qualitative parts of the study are performed independently, and their results are interpreted together [18]. For the quantitative part, FPP participants and certified therapists who provided the program were asked to participate in this study and fill out a questionnaire about their experiences with the program. The qualitative part consisted of five focus groups. Three focus groups were held with FPP participants, one focus group with therapists and one focus group with other stakeholders [15].

### 2.2. Inclusion and Exclusion Criteria

For the focus groups, older adults living in the environment of Amsterdam, Veghel and Best in the Netherlands and participating in the randomized controlled trial investigating the (cost-)effectiveness of the In Balance FPP were invited to participate in this study [15]. Participants were 65 years or older, with an increased risk of falls according to the fall risk screening questionnaire [15,19,20,21]. All participants were able to independently execute activities of daily living (e.g., going to the bathroom and dressing) and walk 100 m. Older adults were included if they were classified as non-frail or pre-frail based on the phenotype concept introduced by Fried et al. [22]. Potential participants were excluded if they participated in a fall prevention intervention in the past 6 months, if they were unable to read and understand Dutch, if they had cognitive impairment defined as a score lower than 19 on the Mini-Mental State Examination [23], or if they had any self-reported uncontrolled comorbid conditions or contraindications for conducting physical exercises during the In Balance intervention (e.g., cardiovascular, neurological and orthopaedic problems). We refer the reader to the description of the study protocol of the (cost-)effectiveness study of the In Balance program for a more elaborate description of the inclusion and exclusion criteria [15]. All therapists who participated in the in the (cost-)effectiveness study were invited to also participate voluntarily in this study [15]. For the stakeholders, we searched for a varied group in our network based on suggestions from the advisory group.

### 2.3. Ethics

Ethical approval was obtained from the Medical Ethical Committee Brabant (project number P2055) on 10 February 2021. This study was conducted according to the principles of the Declaration of Helsinki (7th revision, October 2013) and other guidelines, regulations, and acts such as Good Clinical Practice and the statement conducting research involving humans. All participants of the focus groups signed informed consent and were aware that participation is voluntary. All FPP participants also signed consent for presenting their demographic characteristics. Participants were able to withdraw from the study at any time. In reporting this study, the consolidated criteria for reporting qualitative research (COREQ) were followed [24]. This trial was registered in the Netherlands Trial Register: NL9248 (registered 13 February 2021, URL: https://www.onderzoekmetmensen.nl/en/trial/26195 (accessed on 25 January 2024).

### 2.4. Data Collection 

#### 2.4.1. Questionnaires 

A total of 131 FPP participants were asked, four months after the start of the study (immediately after finishing the FPP), to complete a questionnaire online (by email) or on paper. This questionnaire included questions about the performance of home exercises, satisfaction with In Balance, self-perceived health effects and willingness to pay a financial contribution; we refer the reader to Appendix A for the questionnaire.

All 14 therapists who participated in the randomized controlled trial were asked to fill out a questionnaire about their experiences with the FPP [15]. This questionnaire included satisfaction with the program, the most suitable target group for the program and the costs of the program.

#### 2.4.2. Focus Groups

The duration of each focus group was about two hours. We used baseline characteristics of the FPP participants from data collected in the randomized controlled trial after obtaining informed consent to also use them in this study. The therapists and stakeholders filled out a short questionnaire to collect demographic characteristics. All focus groups were audio-recorded. Audio recordings were saved in a protected online environment and were subsequently deleted from the audio recorder. To create an open environment, an independent female researcher, MM, with over 20 years of experience in conducting interviews and group discussions led the focus groups. MM was not involved with the design of the study and analysis of the results. MvG was present during the focus groups and made field notes but was not actively involved in leading focus group discussions. Participants had no relationship with the independent researcher. This independent researcher developed the discussion manual together with the other authors. Focus groups were organized until data saturation was reached, which was discussed by the research team. Because of privacy considerations, we did not return the transcripts to participants for their review and comments.

### 2.5. FPP Participants 

First, three focus groups were performed with a total of 33 FPP participants. Table 1 includes the interview schedule for this focus group.

### 2.6. Therapists

Second, one focus group was performed with five therapists as part of the randomized controlled trial [15]. This focus group was held online using Zoom Video Communications, Inc., version 5.15.7 (San Jose, CA, USA). Table 2 presents the interview schedule.

### 2.7. Other Stakeholders

Finally, a focus group was held with nine stakeholders. Because representatives from health insurance companies were not able to attend this focus group, they were asked to provide their ideas, thoughts and opinions to the transcript of the focus group. This focus group was held online using Zoom Video Communications, Inc., version 5.15.7 (San Jose, CA, USA). The interview schedule for this focus group is included in Table 3.

### 2.8. Data Analysis

#### 2.8.1. Quantitative Outcomes

Quantitative data were analysed using RStudio version 1.3.1073 (RStudio Team. Boston, MA, USA). Data were described using means and standard deviations for normally distributed continuous variables, medians and interquartile ranges for non-normally distributed continuous variables and numbers and percentages for non-continuous variables.

#### 2.8.2. Focus Groups

The transcription of audio-recorded data was carried out via a transcription service. The data analysis of the focus groups was performed according to the thematic analysis approach of Braun and Clarke [25]. This is an accessible and theoretically flexible approach to identify, analyse and report the patterns and themes within qualitative data [25]. The analysis was carried out using MAXQDA Analytics Pro 2022. MvG was involved in both data collection and (open and manual) coding.

The thematic analysis consisted of 6 different steps. First, researchers DB and MvG familiarized themselves with the data. Transcriptions of the focus groups were independently checked against the audio-recorded data. Transcriptions were read, the researchers made themselves familiar with the data, and initial ideas were written down. Second, the transcript of the first focus group was open-coded by both researchers. The open in vivo codes were discussed, initial axial codes were defined, and a preliminary codebook was developed using different colours for different topics. Subsequently, based on this codebook, DB and MvG open- and axial-coded the transcripts of the other four focus groups independently and discussed them together afterwards. During this process, the codebook was updated and adapted continuously. In the case of inconsistency, the researchers asked for help from a third, independent researcher (BV). Third, via selective coding, themes that had a relation with the aim of the study were identified. Overarching themes were independently identified by comparing and combining the axial codes from step 2. This resulted in a coding tree. Fourth, DB and MvG discussed the overreaching themes and coding tree, which resulted in the pre-final themes. Afterwards, the pre-final themes were presented to and discussed with BV. Fifth, based on this discussion, themes were further defined and named via ongoing reflexive dialogue. The findings were also presented and discussed within the research team. Last, the report was produced; the codes and themes were related to the research question and were interpreted using literature [25].

## 3. Results

### 3.1. Results of the Questionnaires

The mean age of the 104 FPP participants who filled in the questionnaire was 78 (±9.8) years, and most participants were women (n = 87, 68%). Of all FPP participants, 93% were a bit or very satisfied with the program. After following the program, 65% reported perceiving improvements in physical balance. Table 4 presents the results of the FPP participants who returned the process evaluation questionnaire.

Of the 14 therapists who filled in the questionnaire, most were women (79%). Of the therapists, 86% were satisfied with the program and 93% would recommend the program to older adults having difficulties with balance or walking. Deviations from the standard training protocol were reported by 57% of the therapists. The outcomes of the questionnaire are presented in Table 5.

### 3.2. Characteristics of the Participants of the Focus Groups

Table 6 shows the characteristics of the FPP participants, therapists and stakeholders who participated in the focus groups. A number of 33 FPP participants, 5 therapists and 11 stakeholders participated in the focus groups. Most participants in each group were women. The therapists had an average of 18 years of working experience, and the stakeholders had about 10 years of working experience in their current function.

### 3.3. Focus Groups

A total of 838 open codes were collated, formed and divided over sixteen axial codes. These axial codes were combined into six ultimate themes, which are elaborated below. See Table 7 for an overview of the themes with corresponding facilitators.

#### 3.3.1. Recruiting and Motivating Older Adults to Participate 

To optimally benefit from the FPP, it is important that the target group consists of community-dwelling older adults with a fall history or fear of falling. This is the group that is intrinsically motivated to participate in the program:

“*The people who most recognize themselves in things are the people who have had a fall once or twice and very often they say: they didn’t know how that happened, but are still a bit in that shock of: oh, I fell and I have to do something with it. And I think that group is the best to train with.*”[therapist]

Although most eligible participants already experienced a fall, therapists describe that the participants of the In Balance FPP should not be too vulnerable. The appropriate target group can be identified using screening tools that focus on frailty status and fall risk. These screening tools are not generally available yet, but therapists noticed that there is a need for them.

Trial participants in the current study, but also participants who follow the In Balance FPP outside this research, were mostly well-educated older adults with a higher socioeconomic status and without a migration background. Therefore, according to the therapists, most FPP participants are not a reflection of society. To have a more representative group of participants, recruitment should be carried out using a personal approach in the local environment of potential participants: for example, via health professionals, at community centres, or via local newspapers. When recruiting, the social element and the person-centred approach of the program are important to focus on. Also, positive labelling is helpful for recruitment: for example, not using the term ‘fall prevention’ but ‘remaining your balance’ and using language at the ECRL-B1 level to enhance clarity.

#### 3.3.2. Structure and Content of the Program

Overall, FPP participants, therapists and stakeholders were positive about the structure and content of the FPP. Positive aspects were the combination of education and exercises, interdisciplinarity, having the same therapist(s) during the program, and a tailor-made approach:

“*Everyone could do the exercises on their own occasion, at their own pace. I quite liked that.*”[older male participant]

However, several barriers were mentioned. With respect to content, the high number of training sessions was mentioned as a barrier for FPP participants to participate, according to therapists and stakeholders. Second, some FPP participants noticed that more attention should be focused on the mental component after a fall: for example, when fear of falling increases. Such fears may negatively affect self-confidence, and they could be more thoroughly addressed. Furthermore, exercises should focus more on activities of daily living and real situations in practice, and there could be more variations in exercises according to therapists:

“*I notice that I think it’s too bad that: okay, we are practicing this now, but what is it good for in everyday life. Or people, for example, are afraid to ride a bicycle or do certain activities and I also miss the link with that.*”[therapist]

With respect to structure, a consistent and continuous schedule of training sessions is important. Therapists noticed that it is confusing when in the first four weeks, there is one training session per week while the last ten weeks include two training sessions per week. Some therapists mentioned that the number of information meetings is too high. There should also be a regular start of the program, for example, every three months, so potential participants do not have to wait too long before the program starts:

“*You lose a lot of people because for In Balance, you can’t enroll during the course. Then you have organized one and then people have to wait a long time if they are just too late, so that you lose a lot of people again. So actually, you would like some continuity really just every 2 weeks, month, whatever, something somewhere.*”[stakeholder, working at a municipality]

#### 3.3.3. Awareness, Confidence and Physical Effects

A crucial factor for adherence is that FPP participants perceive some degree of training effects. Specific effects that were addressed are more self-consciousness: physical effects, such as balance and strength, and mental effects.

Awareness of fall risk in daily life activities was important according to participants, therapists and stakeholders. Awareness can affect multiple domains: for example, the physical domain, such as being aware of the manner of moving and walking and being aware of one’s own body, strength and posture. But it is also important to be more aware and pay more attention to what someone is doing (e.g., walking the stairs), or that someone is not able to do things as fast as before, and regarding one’s own fall risk and physical activity (e.g., do not walk or do daily tasks in a hurry, but pay attention to it, and plan actions). Moreover, FPP participants also noticed their increased awareness of the environment: for example, making adaptations within and outside of the house and being more attentive to the surroundings:

“*I was even picking things up off the floor in someone else’s house at one point. That is not the intention either, but you’re much more aware of: that can be dangerous.*”[female participant]

FPP participants felt that they had a better physical condition and increased balance and strength, and most participants indicated that they increased their physical activity level after following the FPP. Some FPP participants continued their exercises after the program: for example, standing on their toes when making coffee or brushing their teeth:

“*And I enjoyed doing it, but it actually pushed me to exercise a bit more. I am not the natural gym rat. And this has pushed me to become more of one now and I’m in better shape in that regard.*”[male participant]

Self-perceived confidence increased and fear of falling decreased due to the program:

“*Also when it was a bit slippery and normally you would think like I’m going to be very careful again and now you think I just have to do it like I learned it, that you think, you know, actually it’s going fine.*”[female participant]

#### 3.3.4. Training with Peers

Sharing experiences with peers and learning from each other were perceived as helpful for adherence and wellbeing. FPP participants and therapists indicated that there is a lot of personal attention and participants are willing to help each other. In most groups, social interaction is supported by drinking a cup of coffee together after the training session. Team spirit and its social component were important and provided motivation to come to the training sessions:

“*And I also had a very positive experience. All the things we did. I really liked it and I made some friends, proper lasting friends.*”[female participant]

The therapist played an important role in creating group cohesion and maintaining a good atmosphere. This makes the program more enjoyable and increases adherence. Some FPP participants and therapists mentioned that after the program finished, the group kept seeing each other:

“*I think you suddenly come to the conclusion that you’re 80 years old and within a week so to speak, I almost make new friends.*”[female participant]

Although some of the training sessions took place during a period of COVID-19 with restrictions, the COVID-19 situation did not have noteworthy consequences for the FPP participants.

#### 3.3.5. Funding and Costs

Stakeholders and therapists indicated that paying a contribution increased the motivation and commitment to keep participating in the program. Adherence benefits most from paying for the whole program rather than paying per attended session. However, paying a contribution should not be a barrier to participating in the program. Therefore, it is important that the contribution is not too high so that it remains affordable:

“*We ask for a contribution of 10 euros for the course, which is very little, but is still a difference compared to nothing and we do not want to make a distinction in contributions.*”[stakeholder, innovation and development wellbeing organisation]

A feasible individual contribution for FPP participants should be between EUR 45 and 150 for the complete program according to FPP participants, therapists and stakeholders. The funding differed per municipality where the program was provided, and the contributions also differed between locations. As municipalities are responsible for the funding of local FPPs in the Netherlands, the manner and amount differ between locations. It would be better, according to the FPP participants and therapists, to harmonize their own contributions across the country so that their own financial contributions do not differ between locations.

Also, healthcare insurance companies sometimes financially contributed to the program: for example, when there was an underlying medical condition. It should be made more clear which insurance company contributes which amount of money for FPPs.

#### 3.3.6. Long-Term Continuation

There was little offer in sports or follow-up programs after finishing the 14-week FPP. If there were any, it was locally arranged, and the offer differed per location. There was also a need for the continuation of exercising with the same group as in the FPP:

“*I’ve also had it happen that a dining club was found after the program, where they kept seeing each other. That’s fun too. And this time there was a whole group that wanted to continue training together.*”[therapist]

Securing and trying to pursue a regular start of the In Balance program was suggested:

“*Otherwise, the assurance is of course that you can offer the program regularly so that people also know that: oh yes, you can do that every autumn or every spring and that you know that, with the partners involved: this is a continuous offer here. Just a basic facility that we want to realize.*”[stakeholder, innovation and development wellbeing organisation]

FPP participants noticed that booster sessions should be organized to refresh the information and exercises of the program and determine if more follow-up actions are needed for the FPP participants.

## 4. Discussion

We conducted a process evaluation with FPP participants, therapists and other stakeholders to identify how FPPs can be optimized and further implemented in the community. We found that FPP participants, therapists and stakeholders in general were satisfied with the training program and would recommend it to older adults with an increased fall risk. Moreover, we identified six themes that are important in the optimalisation and implementation of FPPs: (1) recruiting and motivating older adults to participate; (2) structure and content of the program; (3) awareness, confidence and physical effects; (4) training with peers; (5) funding and costs; and (6) long-term continuation. These themes will be further discussed below with respect to the literature and in terms of recommendations.

### 4.1. Recruiting and Motivating Older Adults to Participate

Participants suitable for participation in the FPP are older adults who experienced a fall before and/or have a fear of falling and therefore are intrinsically motivated to participate in an FPP [26]. To recruit a suitable target group, people need to be actively recruited in the local and social domain, and an easily applicable screening tool is needed, as recommended in guidelines for fall prevention [2].

### 4.2. Structure and Content of the Program

As also observed in other research, a combination of education and exercise was experienced as positive [9]. Exercises should focus on the activities of daily living and situations in practice, which may increase adherence to continue carrying out exercises at home [27]. There should be enough variation and different difficulty levels in the exercises to keep participants engaged in the FPP [28]. It was experienced as positive that training could be tailor-made for participants so that every participant can train on their own level while being challenged to maintain effectiveness [29]. Also, training with the same group and the same therapist was mentioned as a facilitator and can provide structure to the FPP. The high number of training sessions was mentioned as a barrier by therapists and stakeholders. However, this intensity of training sessions is one of the core components of effective fall prevention programs and should be maintained to preserve the effectiveness of FPPs [14]. Future programs might consider explaining this in more detail to participants. For individuals who are unable or unwilling to commit to two out-of-house weekly exercise sessions, internet-based interventions, such as StandingTall [11], may serve as an additional viable solution.

### 4.3. Awareness, Confidence and Physical Effects

Self-awareness appeared to be an important aspect, as indicated by FPP participants, therapists and stakeholders in this study. The process of growing awareness and corresponding actions can be divided into three stages: ignoring (continuing a risky activity), gaining insight (realizing the danger in a certain situation), and anticipating (thinking ahead and acting in advance) [30]. Increasing awareness is paired with recognizing one’s fall risk and anticipating situations in daily life. The self-awareness of fall risks is also associated with engagement and motivation in the FPP, because individuals who do not recognize their fall risk or impairments are less likely to appreciate the need and benefits of a fall prevention program [31]. The group intervention may help with the awareness process due to mutual experience sharing; participants observe their own behaviour and the environment around them by exchanging experiences [30].

### 4.4. Training with Peers

Training in a group seemed to be a crucial aspect of the FPP. The social component of the FPP leads to team spirit and willingness to help each other, resulting in more motivation and, subsequently, improved program adherence. This is in line with previous research indicating that social contact is a facilitator for adherence [32]. Participants in group interventions can encourage each other and boost intrinsic motivation to keep exercising [33]. We also found that people learn from each other when sharing experiences, which is important for enhancing self-efficacy [33,34]. Moreover, group exercise has other benefits such as social integration [2,32,33]. A group program can also improve quality of life and health effects [35,36].

### 4.5. Funding and Costs

An overarching topic is a lack of uniformity between different FPP groups, for which funding and costs are an important aspect. Variations existed in funding and individual contributions between locations. More uniformity between groups is recommended so that, for example, an individual’s contribution does not depend on the location where a participant wants to follow the FPP. We found that most older adults, therapists and stakeholders prefer their own contribution when participating in the FPP, because it increases motivation to continue the FPP. However, previous research among older adults showed that 41% of participants indicated an unwillingness or lack of ability to pay for a fall prevention program [35]. This shows that paying a contribution can also be a barrier to participating in an FPP, especially for older adults with a lower socioeconomic status [37].

### 4.6. Long-Term Continuation

The continuation of exercising after the FPP is important for maintaining and increasing long-term effects [38]. A recommendation from the focus groups is to organise follow-up meetings to refresh the information and exercises learned during the FPP. Ideally, there is a smooth transition from an FPP to regular exercise programs or sports activities, such as Tai Chi, walking, walking football, dancing, pilates, or yoga. Possibly, for the continuation of FPPs, eventually, even with the same FPP group, online programs could be used. Such an online program can also help contribute to engagement and socialization: for example, using an age-friendly online platform with a chat or a forum, live sessions on how to do exercises, and videos for review or practice [39]. Digital programs can provide a training program and enhance long-term motivation and continuation at relatively low costs [40]. Organisational factors, such as partnership formation, networking, community capacity, financial resources, a lack of coordination, and the participation of older people, were the most reported influences for the continuation of FPPs, which affect program continuation and sustainability [41,42]. Also, a regular starting period for FPP is important for continuity and provides clarity for the FPP participants and partners involved when the program starts again.

### 4.7. Strengths and Limitations of This Study

A strength of this study is that we had a substantial number of FPP participants, therapists, and stakeholders participating in both quantitative and qualitative studies. The participants were included from several places across the Netherlands; this makes the results more generalizable. Second, all participants were older adults who followed the training program, therapists who provided the program, or stakeholders who are responsible for the offering of FPPs and thus are experts by experience [43]. Experts by experience tend to emphasise experience-related issues, which is their strength and the value-adding reason for involving them. Another strength is that an independent and experienced researcher moderated the focus group and assisted in the design of the focus group guidelines. In doing so, we ought to create a safe and open environment so that participants dare to share their experiences, including negative ones.

Despite efforts to also include people who are less enthusiastic about the In Balance program from the RCT in the focus groups, there may be selection bias. Participants who were positive about the program mostly signed up to participate in the focus groups, which may have led to an overly optimistic view of the In Balance program. Moreover, although we had a substantial number of participants spread out throughout the country, it was quite a homogeneous group of participants; mostly, the group comprised people with a higher socioeconomic status and without a migration background. Ideally, we have had a representative sample of the society in our study. Possibly, people with lower socioeconomic status or a migration background may have other or additional preferences regarding the In Balance program; thus, this may result in other recommendations for the optimalisation of the program. It should also be noted that the questionnaires used in this study are not validated, but they were carefully designed in collaboration with therapists and experts in this field.

### 4.8. Implications and Recommendations

Based on the findings of our study, several recommendations can be made for implementing FPPs. When recruiting and motivating older adults to participate in the FPP, the most eligible target group is community-dwelling older adults with a history of falls and/or fear of falling. Regarding the content and structure of the FPP, the combination of theory and practice and the tailor-made approach was experienced as positive. A crucial factor for adherence is that FPP participants perceive some degree of training effects, such as more awareness. Moreover, sharing experiences with peers and learning from each other were perceived as helpful for adherence and wellbeing. The harmonization of funding and costs across the country is suggested so that paying one’s own financial contribution does not differ for participants between locations. Last, there is a need for the continuation of exercising after the FPP, preferably with the same group as in the FPP.

## 5. Conclusions

This mixed methods study showed that FPP participants, therapists and stakeholders were satisfied with the training program and would recommend it to (other) older adults with a higher fall risk. Six themes were identified that are important for the further optimalisation of the implementation of FPPs: (1) reaching and motivating older adults to participate; (2) structure and content of the program; (3) awareness, confidence and physical effects; (4) training with peers; (5) funding and costs; and (6) long-term continuation. This study provides practical recommendations to optimize and further implement the In Balance program and is also generalizable to other FPPs.

## Figures and Tables

**Table 1 ijerph-21-00162-t001:** Schedule of the focus group with the FPP participants.

Topic Area	Sample Questions
Fall prevention program	What are the positive aspects of the program? What are points of improvement of the program? What is needed to reach more people that will join the program?
Therapist	What did you think of the therapist?
Experiences of self-perceived effects of the program	Did you notice any physical effects? Did you notice any mental effects? Did you notice any social effects?
Financial contribution	How much are you willing to pay for following the program?
Continuation	Did you continue exercising after completing the program?

**Table 2 ijerph-21-00162-t002:** Schedule of the focus group with therapists.

Topic Area	Sample Questions
Target group	Was the training group suitable for the FPP? Why? Which target group do you think is best suited to participate in FPPs? How do you reach the suitable population?
Fall prevention program	What are positive aspects of the program? What are points of improvement of the program?
Effects of the program	Do you think the program has benefited the participants? If so, how?
Implementation	How can we make FPPs more accessible? When we want to roll out the program further, what should we consider?
Financial contribution	Who should bear the costs of FPPs? What can we reasonably ask for a contribution of participants?
Continuation	Are there possibilities for exercising after completion of the program? If so, what is offered? What is needed to encourage people to stay physically active?

**Table 3 ijerph-21-00162-t003:** Schedule of the focus group with the stakeholders.

Topic Area	Sample Questions
Target group	Which target group do you think is best suited to participate in FPPs? How do you reach the suitable population?
Fall prevention programs	What works well in FPP? What does not seem to work as well in FPPs?
Implementation	How can we make FPPs more accessible? When we want to roll out the program further, what should we consider?
Financial contribution	Who should bear the costs of FPPs? What can we reasonably ask for a contribution of participants?
Continuation	How do we keep people moving? What is needed to encourage people to keep moving?

**Table 4 ijerph-21-00162-t004:** Results of the questionnaire of the FPP participants.

	Number of Participants (%)
Satisfaction about the program ^a^	
Very satisfied	77 (75.5%)
A bit satisfied	18 (17.6%)
Neutral	6 (5.9%)
A bit dissatisfied	1 (1.0%)
Very dissatisfied	0 (0.0%)
Would you recommend the program to someone having difficulties with balance or walking ^a^	
Yes	89 (85.6%)
Maybe	14 (13.5%)
No	1 (1.0%)
Do you experience your balance to be improved after following the program? ^a^	
Improved strongly	13 (12.6%)
Improved a little	54 (52.4%)
Remained the same	33 (32.0%)
Decreased a little	3 (2.9%)
Decreased strongly	0 (0.0%)
Did COVID-19 play a role in your choice to participate in the fall prevention program? ^a^	
Yes, I wanted more physical and social activity again	10 (9.6%)
Yes, I was unsure about group activities	0 (0.0%)
Yes, because of another reason	2 (1.9%)
No	94 (90.4%)
Did you do the exercises yourself at home? ^a^	
Yes	86 (19.7%)
Multiple times per week	59 (69.4%)
Once a week	23 (27.1%)
Less than once a week	3 (3.5%)
No	18 (4.1%)
Doing the exercises at home took me too much time	5 (4.8%)
Due to health problems	0 (0.0%)
I could not bring myself to do it	14 (13.5%)
I didn’t have a suitable room in which I could do the exercises	0 (0.0%)
I found the exercises too difficult/dangerous	0 (0.0%)
I found the exercises too tiring	2 (1.9%)
If you were asked to contribute for your participation in the fall prevention program, how much would you be willing to pay? ^a^	
I would not pay a personal contribution	20 (19.4%)
1 to 25 euros	22 (21.4%)
25 to 50 euros	34 (33.0%)
50 to 75 euros	14 (13.6%)
75 to 100 euros	11 (10.7%)
More than 100 euros	2 (1.9%)

^a^ Presented as n (percentage).

**Table 5 ijerph-21-00162-t005:** Results of the questionnaire of the therapists who provided the fall prevention program.

	Number of Participants (%)
Satisfaction about the program ^a^	
Satisfied	12 (85.7%)
Neutral	2 (14.3%)
Dissatisfied	0 (0%)
Deviated from standard training program (yes) ^a^	8 (57.1%)
Would you recommend the program to someone having difficulties with balance or walking (yes) ^a^	13 (92.9%)
Did you find the participants to whom you gave the program suitable for participation (yes)? ^a^	9 (64.3%)
Did COVID-19 play a role in your choice to participate as a therapist? ^a^	
Yes, group size has been adjusted	1 (7.1%)
Yes, I made some adjustments in providing the program	1 (7.1%)
Yes, I was unsure about group activities	1 (7.1%)
No	11 (78.6%)
If participants were asked to contribute for their participation in the program, which amount do you think this contribution should be? ^a^	
0 to 25 euros	0 (0%)
25 to 50 euros	2 (14.3%)
50 to 75 euros	4 (28.6%)
75 to 100 euros	2 (14.3%)
More than 100 euros	5 (35.7%)

^a^ Presented as n (percentage).

**Table 6 ijerph-21-00162-t006:** Characteristics of the older adults receiving the FPP who participated in the focus groups.

	Number of Participants (%)
Age (years) ^a^	
FPP participants	76.7 (SD 5.7)
Therapists	41.6 (SD 17.5)
Stakeholders	50 (SD 20.1)
Sex (female) ^b^	
FPP participants	21 (63.6%)
Therapists	4 (80%)
Stakeholders	8 (72.7%)
Fall history of the FPP participants ^b^	
Never	9 (27.3%)
One time	12 (36.4%)
Two times	7 (21.2%)
Three or more times	5 (15.2%)
SPPB score of the FPP participants ^a^	9.8 (SD 1.5)
Educational level ^b^	
Primary school	0 (0%)
Equivalent to junior high school, apprenticeship	9 (27.3%)
High school, college, university	24 (72.7%)
Number of years of working experience in function ^a^	
Therapists	18.2 (SD 15.4)
Stakeholders	9.9 (SD 13.2)
Therapists ^a^	
Number of years certified as In Balance therapist	4.8 (SD 1.6)
Total number of In Balance programs provided	5.8 (SD.4)

^a^ Presented as mean (standard deviation, SD); ^b^ Presented as n (percentage). SPPB = Short Physical Performance Battery.

**Table 7 ijerph-21-00162-t007:** Overview of the themes with corresponding facilitators for optimalisation and implementation of the FPP.

Theme	Facilitator
Recruiting and motivating older adults to participate	Addressing an eligible target group: (a) Community-dwelling older adults with a fall history or fear of falling (b) Intrinsically motivated (c) Should not be too vulnerable Screening tools which therapists can use for screening of potential participants for the criteria mentioned above under ‘Target group’ Recruitment via personal approach with a focus on: (a) Social element (b) Person-centred approach (c) Positive labelling Use of Language at the ECRL-B1 level during recruitment and training program
Content and Structure of the program	5. Content (a) Combination of theory (education) and practice (exercises) (b) Attention for the mental consequences after a fall (c) Exercises with a focus on activities of daily living and situations in practice (d) Enough variation in the exercises (e) A limited number of information meetings 6. Structure (a) A tailor-made approach (b) Having the same therapist(s) during the program (c) Interdisciplinarity of the program (d) Regular and continuous offer of the program (e) A consistent and continuous schedule of the training sessions with a training program
Awareness, confidence and physical effects	7. The perception that the program is effective with respect to: (a) Awareness (b) Physical effects (c) Mental effects
Training with peers	8. Sharing experiences and learning from each other 9. Personal attention 10. Willingness to help each other 11. Possibility to socialize during and after the program 12. Team spirit 13. Social component 14. Therapist is important in maintaining a good atmosphere
Funding and costs	15. Small financial contribution to participate 16. Equalizing funding across the country 17. More clarity about contribution of insurance companies
Continuation after the fall prevention program	18. Enough and varied offer in follow-up programs or sports activities 19. An option to continue training with the same group 20. Follow-up meetings to refresh information and exercises

## Data Availability

Only the researchers have access to audio recordings and transcripts due to the individual privacy considerations of participants.

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
