# Peer review of "Optimizing and Implementing a Community-Based Group Fall Prevention Program: A Mixed Methods Study"

_ijerph, 2024, doi:10.3390/ijerph21020162_

Round 1
Reviewer 1 Report
Comments and Suggestions for Authors
The article is interesting and contributes to improving scientific and professional knowledge regarding the topic under analysis.
In the introduction / framework, I consider that it should be more contextualized and based on equivalent national and international experiences. This article does not highlight this knowledge already acquired. It makes reference to other work by the team of researchers, which would be avoided given that future contributions will still be made.
for example: on line 50, it would be important to have a more recent reference instead of the revised one (number 9).
It would be important in the introduction (line 63) to give a broader and more comprehensive view of the work being carried out beyond that of the authors themselves.
In Methods (lines 69 to 78), the description of the quantitative and qualitative part of the study (methods) is better to be described after the selection of the participants.
It will be good for understanding the study to report the moment were questionnaires were made. (moment in the program), In table 4, the data is presented in wrong lines in "Do you experience your balance to be improved after following" and "Did you do the exercises yourself at home?".
good work
Author Response
Thank you for giving us the opportunity to revise our manuscript entitled ‘Optimizing and implementing a community-based group fall prevention program: a mixed methods study’ and the willingness to consider our manuscript for publication in International Journal of Environmental Research and Public Health. We appreciate the fast review process concerning our manuscript. Please find our responses to the reviewers' comments below. The phrases in italics are changed or added in the revised manuscript. We want to thank the reviewers for their comments which improved our manuscript. We hope that the revised manuscript is acceptable for publication.
General comment: The article is interesting and contributes to improving scientific and professional knowledge regarding the topic under analysis.
Response: Thank you for your comment. We appreciate your positive feedback.
Comment 1: In the introduction / framework, I consider that it should be more contextualized and based on equivalent national and international experiences. This article does not highlight this knowledge already acquired. It makes reference to other work by the team of researchers, which would be avoided given that future contributions will still be made.for example: on line 50, it would be important to have a more recent reference instead of the revised one (number 9).
Response and action: We added an additional paragraph in the introduction section to provide more context and references on (national and international) fall prevention programs, see lines 38-57. We specifically referred to the study of Faber et al. from 2006, because this has been the only study ever done to on the effectiveness of the widely implemented In Balance fall prevention program in the Netherlands. One of the reasons why we are re-evaluating the (cost-)effectiveness and optimizing the implementation of the In Balance fall prevention program, is that since the study of Faber et al., the program has underwent several changes due to the implementation in clinical practice. Moreover, since fall prevention programs are difficult to implement in the community, our study can help by providing generic knowledge that is important for other fall prevention programs, developers of fall prevention programs and policy makers [1]. We also highlighted this in the revised introduction.
Comment 2: It would be important in the introduction (line 63) to give a broader and more comprehensive view of the work being carried out beyond that of the authors themselves.
Response and action: As stated above, we added an additional paragraph in the introduction section to provide more context on fall prevention programs that have been implemented (inter)nationally, see lines 38-57.
Comment 3: In Methods (lines 69 to 78), the description of the quantitative and qualitative part of the study (methods) is better to be described after the selection of the participants.
Response and action: The description of the quantitative and qualitative part of the study (methods) is now moved after the description of the selection of the participants, see lines 89-102.
Comment 4: It will be good for understanding the study to report the moment were questionnaires were made. (moment in the program)
Response and action: The questionnaires were administered four months after start of the study (immediately after finishing the fall prevention program), see lines 139-140.
Comment 5: In table 4, the data is presented in wrong lines in "Do you experience your balance to be improved after following" and "Did you do the exercises yourself at home?".
Response and action: Indeed, the data was presented in wrong lines. Now the data is presented correctly.
General comment: good work.
Response: Thank you.
On behalf of all authors,
Yours sincerely,
Maaike van Gameren
Reviewer 2 Report
Comments and Suggestions for Authors
The manuscript is generally well-written.
In Table 7, could you please clarify the distinction between points marked with solid and open circles?
It would be beneficial if the authors could include additional information such as the participants' fall history and some indicators of physical function, such as walking speed and balance assessments if available. Moreover, as participant characteristics mentioned as a limitations, including data on socioeconomic status and the percentage of participants with a migration background, would enhance the comprehensiveness of the study.
I noticed that one of the negative feedback points regarding 'the high number of training sessions' has not been addressed in the discussion. It would be helpful if the authors could provide some insights or suggestions to address this feedback.
Author Response
Thank you for giving us the opportunity to revise our manuscript entitled ‘Optimizing and implementing a community-based group fall prevention program: a mixed methods study’ and the willingness to consider our manuscript for publication in International Journal of Environmental Research and Public Health. We appreciate the fast review process concerning our manuscript. Please find our responses to the reviewers' comments below. The phrases in italics are changed or added in the revised manuscript. We want to thank the reviewers for their comments which improved our manuscript. We hope that the revised manuscript is acceptable for publication.
General comment: The manuscript is generally well-written.
Response: Thank you for this compliment.
Comment 1: In Table 7, could you please clarify the distinction between points marked with solid and open circles?
Response and action: We agree that the format of the table may have been a bit confusing. We changed the solid and open circles to respectively 1, 2, 3 and a), b), c), etc. to clarify that what previously were the solid circles (1, 2, 3, etc.) indicated facilitators and the open circles (a, b, c, etc.) indicated elements of the facilitator, see Table 7.
Comment 2: It would be beneficial if the authors could include additional information such as the participants' fall history and some indicators of physical function, such as walking speed and balance assessments if available.
Response and action: Thank you for your suggestion. We added the participants' fall history and their SPPB scores as a measure for functional status of the participants in Table 6.
Comment 3: Moreover, as participant characteristics mentioned as a limitations, including data on socioeconomic status and the percentage of participants with a migration background, would enhance the comprehensiveness of the study.
Response and action: We agree that reporting socioeconomic status will increase the interpretation of our findings. We therefore added, as a measure of socioeconomic status, participants’ education level in Table 6. Unfortunately, we do not have data about migration status.
Comment 4: I noticed that one of the negative feedback points regarding 'the high number of training sessions' has not been addressed in the discussion. It would be helpful if the authors could provide some insights or suggestions to address this feedback.
Response and action: Thank you for noticing this. We added in the discussion the following sentence: ‘The high number of training sessions was mentioned as a barrier by therapists and stakeholders. However, this intensity of training sessions is one of the core components of effective fall prevention programs and should be maintained to preserve the effectiveness of FPPs [2]. For individuals who are unable or unwilling to commit to two out of house weekly exercise sessions, internet based interventions, such as StandingTall [3], may serve as an additional viable solution.’, see lines 406-412.
Thank you for your consideration. We are looking forward to your response.
On behalf of all authors,
Yours sincerely,
Maaike van Gameren
Reviewer 3 Report
Comments and Suggestions for Authors
Dear all, here are some considerations to complement your study:
Introduction
1. I suggest the presentation of different FPPs around the world, as well as their main characteristics;
2. From this, I suggest that the programs are compared, as well as their limitations;
3. To date, the study still lacks a consistent justification, which can be constructed based on a better review of the current literature.
Methodology
Questionnaires
1. Have the questionnaires answered by participants and therapists been previously validated? How were they built? If there is no information about this, I suggest presenting it as a strong limitation of the study;
2. I suggest better clarification on "mixed study" with a theoretical framework. This is the methodological basis of the paper;
3. I suggest presenting inclusion criteria used to select participants and therapists: this is fundamental in a study;
Comments on the Quality of English LanguageOk
Author Response
Thank you for giving us the opportunity to revise our manuscript entitled ‘Optimizing and implementing a community-based group fall prevention program: a mixed methods study’ and the willingness to consider our manuscript for publication in International Journal of Environmental Research and Public Health. We appreciate the fast review process concerning our manuscript. Please find our responses to the reviewers' comments below. The phrases in italics are changed or added in the revised manuscript. We want to thank the reviewers for their comments which improved our manuscript. We hope that the revised manuscript is acceptable for publication.
Dear all, here are some considerations to complement your study:
Introduction
Comment 1: I suggest the presentation of different FPPs around the world, as well as their main characteristics;
Response and action: We added an additional paragraph in the introduction section to provide more context on (national and international) fall prevention programs, see lines 38-57. We provided an overview of the Otago, StandingTall and Stepping On fall prevention programs. See: An example of such a FPP is the Otago Exercise Program (OEP) for community-dwelling older adults who can individually exercise safely [4]. This is a globally implemented intervention that contains individually-tailored strength-, balance- and mobility exercises [4]. These exercises last 30 minutes, participants must exercise three times a week and are encouraged to walk outside at least twice a week [4].
Another example is the home-based eHealth intervention StandingTall [3]. This intervention consists of a balance training by individually-tailored exercise prescription to improve balance by a tablet at home [3]. Participants are encouraged to exercise at least 120 min per week for 2 years [3]. A third FFP example is the Stepping On program which aims to improve self-efficacy and include exercises focusing on improving balance, strength, and encourages behaviour that stimulates safety and mobility (such as emphasizing the influence of medication on the risk of falls and practicing mobility techniques in an outdoor location) [5]. Stepping On includes weekly two hour sessions conducted for seven weeks [5].
Comment 2: From this, I suggest that the programs are compared, as well as their limitations;
Response and action: In the abovementioned additional paragraph, we compared the Otago, StandingTall and Stepping On fall prevention programs and their specific characteristics with each other to show similarities and differences in the programs and target groups. Hopefully, readers will get a better overview of existing (and implemented) fall prevention interventions.
Comment 3: To date, the study still lacks a consistent justification, which can be constructed based on a better review of the current literature.
We hope that the additional paragraph in the introduction (see lines 38-57) provides more justification for conducting the study. Moreover, we added the sentence ‘Additionally, results of this study can provide new insights for the implementation of other FPPs in a real life setting [1].’ in lines 83-84. We expect the justification for the study to be more clear now.
Methodology
Questionnaires
Comment 4: Have the questionnaires answered by participants and therapists been previously validated? How were they built? If there is no information about this, I suggest presenting it as a strong limitation of the study;
Response: Thank you for your comment. As we designed the questionnaires for the participants and therapists ourselves, these have not been previously validated. However, therapists and other experts in the field were involved in the design of the questionnaires. Therefore, we did not consider the fact that the questionnaires were not validated as a limitation of this study. However, we added the following to the discussion: ‘It also should be noted that the questionnaires used in this study were not validated, yet they were carefully designed in collaboration with therapists and experts in this field.’, see lines 484-486.
Comment 5: I suggest better clarification on "mixed study" with a theoretical framework. This is the methodological basis of the paper;
Response and action: We now more specifically explained the convergent parallel mixed methods design as a theoretical framework for this study, see lines 95-97: ‘For the current study, we used a convergent parallel mixed methods design. This design includes that the quantitative and qualitative parts of the study are performed independently, and their results are interpreted together [6].’
Comment 6: I suggest presenting inclusion criteria used to select participants and therapists: this is fundamental in a study;
Response and action: Thank you for your comment. We added inclusion and exclusion criteria of the older adults, therapists and stakeholders in the methods section, see lines 104-123.
Thank you for your consideration. We are looking forward to your response.
On behalf of all authors,
Yours sincerely,
Maaike van Gameren
Round 2
Reviewer 1 Report
Comments and Suggestions for Authors
The modifications are an improvement.
Reviewer 3 Report
Comments and Suggestions for Authors
Dear authors, I consider that the study was extensively worked on. Therefore, I suggest publishing it.
Yours sincerely
Reviewer